# The Potential Impact of a Dog Training Program on the Animal Adoptions in an Italian Shelter

**DOI:** 10.3390/ani12141759

**Published:** 2022-07-08

**Authors:** Danila d’Angelo, Luigi Sacchettino, Angelo Quaranta, Michele Visone, Luigi Avallone, Claudia Gatta, Francesco Napolitano

**Affiliations:** 1Department of Veterinary Medicine and Animal Production, University of Naples Federico II, 80137 Naples, Italy; danila.dangelo@unina.it (D.d.); sacchettinoluigi@gmail.com (L.S.); avallone@unina.it (L.A.); 2Animal Physiology and Behavior Unit, Department of Veterinary Medicine, University of Bari Aldo Moro, 70121 Bari, Italy; angelo.quaranta@uniba.it; 3Dog Park, 80044 Naples, Italy; m.visone@caniledogpark.com; 4CEINGE-Biotecnologie Avanzate Franco Salvatore s.c. a r.l., 80145 Naples, Italy

**Keywords:** training program, shelter dog, dog adoption, dog behavior, positive reinforcement

## Abstract

**Simple Summary:**

Human–dog relationships are mainly focused on the physical and emotional wellbeing, and has much evolved in the last decades, becoming even more intense over time. Such a novel conception should also be applied to the life that takes place in kennels, which have been mistakenly regarded as a landfill or a burden on society for too many years. Here, we took into consideration an Italian shelter, and analyzed 555 adopted dogs who underwent a well-detailed behavioral training program, to assess the potential impact of the education upon the adoption of attitude. We documented a higher increase in the number of adoptions both for adult and senior animals when compared to the age-matched untrained dogs. Collectively, our data highlight the importance of a proper training, mainly accomplished with reward methods, to get a more suitable and balanced owner-dog attachment.

**Abstract:**

One of the main concerns of the human–dog relationship is today associated with the quality life inside the kennels, which are very often regarded as animal dump where dogs are exiled, representing a burden on society. In the present study we sought to investigate the importance of performing an appropriate behavioral program on the adoption chances within an Italian shelter, near Naples (Ottaviano). In this respect, we enrolled 555 adopted dogs of different ages, who followed a tailored-4-month lasting training program between 2018 and 2020. Once entered there, they were carefully examined by the veterinary behaviorist, and directed towards a suited training program, to improve living conditions. We documented a higher number of both adult and senior dogs who left the kennel and were adopted, compared to the age-matched untrained animals (*n* = 479), housed in the same kennel from 2015 to 2017. Taken together, the present data highlight an important role for training in improving the natural attitudes of the companion dogs, thus pointing towards a better human–animal bond.

## 1. Introduction

Animals are involved in a variety of human life aspects, ranging from farming and servicing to research and companionship. Notably, human–animal bonds have significantly evolved, so pets are not only employed for work, house protection and hunting mice, but nowadays they also have some social action, and provide a fruitful friendship to humans [1]. On the other hand, given their ability to feel both positive and negative emotions, animals can even benefit from humans, thus highlighting the importance of developing practical welfare assessment indicators either for husbandry or pets [2]. Despite the deep connection between humans and dogs, there are multiple dysfunctional issues arising in these animals to take into account, including excessive interspecies aggression [3], fear and anxiety, or abnormal repetitive behaviors [4], thereby becoming victims of anthropomorphism, animal hoarding [5], and eventually ending up in kennels, which very often are regarded as animal dumps, a place where they spend the rest of their life in exile. Given that the killing of kenneled dogs is not allowed in Italy, unless they are suffering from untreatable diseases or proven to be extremely dangerous to human health [6], such a no-kill policy can cause the shelters to be very generally thronged and represents a burden on society [7]. Indeed, social and spatial restrictions generally bring about an increased excitement, aggression, and uncertainty in dogs, thereby making them generally less suitable to be adopted [8,9,10]. On the other hand, dog training and, even more, socialization programs have already been turned out to be very successful, since they improve the quality of life of sheltered dogs and allow them to establish close relationships with humans [11,12,13]. Moreover, sociability might predict the adoptability level for kenneled dogs, most likely because adopters believe that animals who display social behavior are more friendly and less aggressive [14]. While there is still little knowledge of whether and how social interaction may affect adoption success, dogs that normally behave withdrawn, asocial, or frightened cut down their chances to be adopted [15,16]. In addition, dogs who ignore potential adopters when they tried to start playing were less prone to be adopted, thus suggesting that engagement in playtime with potential owners and proximity to them appeared to increase the likelihood of adoption [17]. Thus, given the importance of the positive reinforcement on physical and mental health, several studies about this topic, rather than dogs [18], have also been addressed in other species, such as horses [19]. Moreover, Vieira de Castro and colleagues reported that aversive stimuli lead to high stress-related behaviors in dogs, showing higher cortisol levels even after carrying out the training, and caused them to be more “pessimistic” in a cognitive bias task [20]. However, it is worth underlining that the best quality of life in shelter dogs can be ensured if, alongside validated tools aimed at improving an overall welfare, a proper behavioral training program is taken into account [21]. Indeed, in a previous study, carried out on 202 kenneled dogs from thirteen Dogs Trust Rehoming Centers, the authors documented that animals housed in quiet, furnished kennels with free access to enrichment enjoyed a better lifestyle than dogs housed in noisy kennels [22]. Therefore, based on these notions, in the present work we tried to address the potential impact of the educational positive reinforcement training upon behavioral functions in kenneled dogs, thus potentiating the attitude to establish a more suitable human–dog interaction. To this aim, we analyzed 1034 adopted, trained and non-trained dogs, housed in one of the most innovative shelters in southern Italy, between 2015 and 2020.

## 2. Materials and Methods

### 2.1. Ethics Statement

No special permission for use of dogs in such behavioral studies is required in Italy, since dogs were observed during their daily life and within their familiar shelter pens. We sent a formal request to the manager and veterinarian of the shelter dog in Ottaviano, who gave us the permission. All procedures were performed in full accordance with Italian legal regulations.

### 2.2. Dog Kennel

The building is in the Municipality of Ottaviano, a city with a highly dense populations near Naples, Southern Italy, spanning on an area of about 16,000 square meters. It has about 300 boxes, and the most organized in multiple boxes to meet the social needs of the canine species. Further details of the kennel have already been described in d’Angelo et al., 2021 [23]. Each box presents a surface of 6 square meters/dog, with an open space in front of it, where animals can have a walk and interact each other, and equipped with a sound system, which broadcasts classical music [24]. The veterinary part occupies an area of approximately 200 square meters, and includes the Operating Room, Outpatient Clinic, Radiology, Analysis Laboratory, and Hospitalization with high technological innovation monitoring.

### 2.3. Animals Enrolled in the Study

In the present work we documented the activity of 1034 mixed-bred dogs (488 males and 546 females), hosted in the kennel of Ottaviano, and adopted between 2015 and 2020. Out of 1387 untrained dogs who entered the kennel between 2015 and 2017, 479 were adopted in such a time window. The behavioral program started off from 2018, as reported in the next paragraph, and engaged 735 dogs, 555 of those were adopted. Depending on their age, the dogs were grouped into four categories, namely juvenile (7–11 months); young-adult (1–2 years); adult (3–8 years); senior (9–17 years) [25].

### 2.4. Training Program

Dogs that we analyzed in the present work underwent a careful behavioral assessment and categorization by veterinary behaviorists, using the evaluation scale for emotional disturbances of dogs (EDED Scale) of Pageat [26], which allows the classification of dogs’ behavior, according to the presence/absence of centripetal and centrifugal behavior, and the expression of homeostasis or emotional disturbances. The centripetal activities were represented by feeding, drinking, self-directed behaviors and sleep, while the centrifugal ones relied on social contacts, exploratory capacity and aggression. For every single behavior considered, a specific score was achieved to the subjects that, once added together, allowed us to get a final score, and to draw general emotional state in each animal. The dogs selected in the present study had a score from 9 to 12, corresponding to a normal state for emotional and cognitive profiles. After the behavioral visit, the dogs hosted were followed by dog trainers or trainer students during their internship, under the supervision of a tutor (expert dog trainer). The training lasted for 4 months and was performed once a week (on average), using positive reinforcements (treats, vocal and caress, play together) and gentle management (no physical or psychological pressure was induced). Each session approximately lasted 60 min, with a break of 15 min after 30 min of working to safeguard animal welfare. At least two activities were carried out in each session. Each session was preceded by the previous learned activity to assess cognition abilities (Table 1).

### 2.5. Statistical Analysis

Statistical analysis was performed with GraphPad (version 9.0; La Jolla, CA, USA). The number of dog adoptions based on their age was assessed using one-way ANOVA, followed by Holm–Šídák’s multiple comparisons test. The effect of training upon dog adoptions was analyzed through the unpaired Student’s *t* test. The evaluation of adopted vs. non-adopted dogs was carried out by means of Chi-squared test. Results were considered statistically significant for *p* < 0.05.

## 3. Results

### 3.1. Age-Dependent Dog Adoptions of the Analyzed Kennel

Here, we investigated the age-dependent effect upon dog adoptions, from 2015 to 2020 (Table 2).

In this respect, we focused on four groups of age: juvenile: 7 to 11 months; young-adult: 12–24 months; adult: 3–8 years; senior: 9–17 years. We firstly evaluated whether our experimental data even fit to a normal distribution. Accordingly, the Shapiro–Wilk normality test showed that, within each analyzed group, values were normally distributed (juvenile: W = 0.9119, *p* = 0.4488; young-adult: W = 0.9638, *p* = 0.8483; adult: W = 0.9673; *p* = 0.8737; senior: W = 0.9468; *p* = 0.7147). Then, one-way ANOVA showed an overall significant effect of the dogs age over the number of adoptions (F_(3,20)_ = 21.32 *p* < 0.0001). In particular, we documented a higher number of younger dog adoptions when compared to the aged ones (Holm-Šídák’s comparisons test; juvenile vs. young-adult *p* = 0.0008; juvenile vs. adult: *p* < 0.0001; juvenile vs. senior: *p* < 0.0001) (Figure 1).

### 3.2. Impact of the Training upon Dog Adoptions

Then, in order to assess the effect of training on the efficacy of dog adoptions, we analyzed two different groups od adopted animals, one including dogs housed from 2015 to 2017, not enrolled in the training program, and another one, consisting of dogs housed from 2018 to 2020, who had been undergoing a 4-mounth-training program (Table 1). Statistical analysis, carried out within each age considered, showed no main effect of training on the number of adoptions in juvenile and young-adult groups (*p* > 0.05, Student’s *t* test). On the other hand, we found a higher number of adoptions in both adult and senior trained dogs, compared to age-matched animals who were not trained (adult; *p* = 0.0324; senior; *p* = 0.0351), as shown in Figure 2. Finally, when we evaluated the influence of the size upon the train-dependent adoptions we failed to find any significant effect (two-way ANOVA, treatment x size interaction: F_(3,16)_ = 1.387, *p* = 0.2828, data not shown).

## 4. Discussion

The dogs analyzed in the present work were selected from the guests of the Coop kennel, Dog Park of Ottaviano (Naples, Italy), where they were housed after being caught on the territory, by the local health authority. Our data point towards a potential impact of educational training on dog adoption. Once they arrived at the shelter the analyzed dogs were suddenly subjected to a clinical examination and laboratory testing (routine biochemical and hematological evaluations), and behavioral characterization as well, to certify their general health status. Then, they were grouped into different categories, tailored to behavioral attitudes, in order to shorten housing time in the kennel, and to improve the chances of adoption [11,27]. In addition, in line with work by Jensen et al. (2020), indicating that about 25% of the animals are relinquished because of their behavioral issues, dogs recovered from behavioral disorders might be easily adopted [28]. In the present work we documented a significantly higher number of total adoptions in the trained dogs, compared to the control animals, since out of over 735 trained dogs, 555 were adopted, instead of 479 out of 1387 non-trained animals (Chi-squared test: df = 322.9, 1; z = 17,97; *p* < 0.0001). This apparently amazing data should be considered with caution, since adoption is a complex phenomenon, affected by many factors, besides training per se, including appearance, social interaction with the adopter and personality [29,30], as well as the need for humans to improve their social and emotional wellbeing. Accordingly, although there is no significant year effect between dogs’ age and 2015–2020 period (One-way ANOVA: F_(5,18)_ = 0.2465, *p* = 0.9362), our data showed an overall increase of adoptions in 2020, compared to the previous years, thus suggesting that, under stressful situations, such as that experienced during the forced lockdown, caused by SARS-CoV2 pandemic, humans might require to establish an emotional osmosis with companion animals to get pleasure from them [31]. Thus, to give a reliable support to our findings, more accurate studies, which deal with as many adoption-related factors as possible, are mandatory. Health-related issues of either dogs or adopters, a lack of time and human deprivation often urge the owner to surrender the animals to the kennel [28,32,33,34,35,36]. A further key factor to achieve a conscious choice is based on the expectations of the adopters towards dog ownership, prior to adoption itself and their gained experience with dog behavior. Powell and coworkers documented that owners who returned their dog to the shelter within three months from the adoption had higher expectations of their dog and human–dog relationship, although animals exhibited a non-dysfunctional behavior when compared with adopters who did not return their dogs [37]. The same authors found that two-thirds of owners experienced dog behavioral concerns following adoption, even though some of them reduced over time (e.g., difficulties with training and fear). Noteworthy, findings from Shore’s group reported that about 50% of relinquishing adopters considered the return process “hardly likely”, and 41% indicated that they would not adopt one more animal in the future, while 13% weren’t sure if they would do it again [38]. The unsuccessful animal adoptions may detrimentally affect the desire of people who are looking for to own a companion animal in the future. Therefore, trying to minimize the huge variability associated to the dog adoption, we used a biased approach, focusing on the potential effect of the training upon adoption chances in an age-related manner. Our data substantiated a significant increase of the adoption numbers, in both adult and senior trained dogs, housed from 2018 to 2020, compared to the age-matched untrained animals, who entered the kennel between 2015 and 2017 (see Figure 2). Nevertheless, we argue that the presence of a qualified dog trainer and vet behaviorist in the kennel, who oversees the categorization of the behavioral profile, might allow a more suitable adoption choice by the owner. Therefore, the combined effect of the educational training with a higher number of interspecies contacts likely let the animals increase their wellbeing [39]. Accordingly, the kennel in Ottaviano is equipped with a diffusion system of classical music, playing every day from 2 to 7 p.m., thereby increasing resting behavior and decreasing distress as well. Such an innovative approach allows us, on one hand, to detect the strengths and weaknesses of each dog and, on the other, make shelter dogs more attractive for potential adopters from a behavioral perspective, thus tackling the overall expectations. In this line, Weiss et al. (2012) reported that most adopters gave importance to the information about the animal’s health and behavior from a staff member or volunteer, rather than cage cards [30]. Again, several studies suggested that training programs, which are regarded as one of the main enrichment plans, might enhance desirable behaviors and decrease unwanted ones in shelter dogs that eventually can improve their welfare [40]. Indeed, Wells and colleagues emphasized the notion that dog’s behavior is much more important to a potential adopter than the canine’s physical appearance [41]. Therefore, training programs pave the way for predictable interactions, increase the management of the environment and improve the lifestyle of dogs [11,42]. In this context, the strategy of the kennel we considered in the present work relies on a detailed 4-month-lasting program to obtain better housing conditions of dogs, by shaping their emotional and cognitive skills. We documented, for the first time, the positive impact of the behavioral training on the ability of dogs to be adopted, in an age-dependent manner, since the number of adoptions was significantly higher for adult and senior dogs who were enrolled in the educational program when compared to the age-matched controls. Puppies in the kennel are generally taken into account by people, since they normally have more chances to be adopted than adults and seniors [29,43], most likely because their characteristics and disposition might change over time as they mature. On the other hand, a senior dog is less likely to change because his personality has already developed. Accordingly, our data collected between 2015 and 2020 showed that there is a significant age effect, since juvenile dogs were more easily adopted when compared to the older animals. Svoboda and colleagues reported that 10 to 12 year-old dogs are more likely to be socially isolated, thus ending up being euthanized [44], because they generally have to cope with several age-related dysfunctions, and become a burden on owners in terms of daily life management. Indeed, alongside medical issues arising over time, including cardiovascular diseases, renal failure, joint fractures and poor mobility [45], a not-negligible percentage of aged dogs is relinquished because they very often develop emotional and cognitive problems, associated with aggressiveness, excessive vocalization, disobedience, biting, escape or hyperactivity [46]. Interestingly, although it is still an initial study, we might reasonably state that the educational program carried out in the present work produced a beneficial effect on the older dogs, who had more chances to be adopted than the age-matched untrained animals. Taken together, we can hypothesize that a tightly scheduled and tailored training program might “shape” emotional and cognitive abilities in dogs, and represents a key factor to the reinstatement of a better human–dog relationship. However, further studies on a wider cohort of dogs, sorting by sex, breed and uniform background, are mandatory to better address this issue.

## 5. Conclusions

Here, we documented the beneficial effects of the training program upon the adoption rate of dogs housed in a southern Italy kennel, in an age-dependent manner, since adult and senior trained dogs were more suitable to be adopted, thus suggesting that, despite the multifactorial issues related to dog adoption, managing in a proper way the dysfunctional aspects experienced by dogs, including excessive aggression, fear and anxiety and abnormal repetitive behaviors, may lay the groundwork to ensure, on one hand, the most suitable life conditions for the hosted dogs and, on the other, a substantial harmonization for human–dog relationships. In this respect, behavioral categorization processes should be taken into account to allow a better dog–human match, preserving the needs of both of them, and improving human awareness in choosing the “right” dog.

## Figures and Tables

**Figure 1 animals-12-01759-f001:**
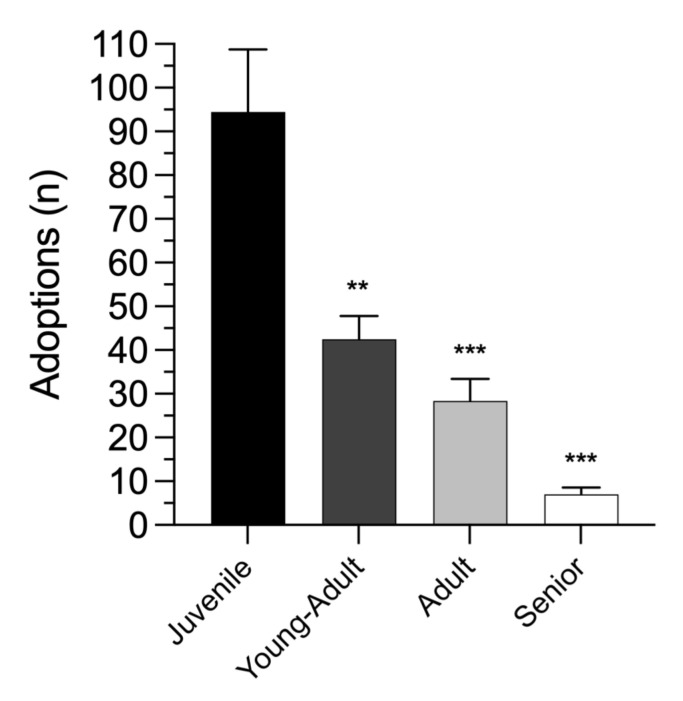
Age-dependent dog adoptions in the analyzed shelter, from 2015 to 2020. ** *p* < 0.01, *** *p* < 0.0001, compared to juvenile group (one-way ANOVA, followed by Holm-Šídák’s multiple comparisons test). All values are expressed as mean ± SEM.

**Figure 2 animals-12-01759-f002:**
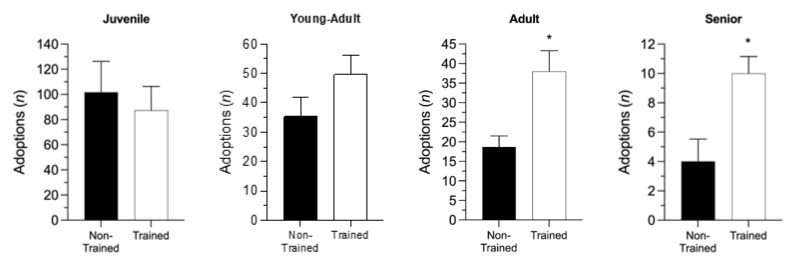
Effect of training upon age-dependent dog adoptions. * *p* < 0.05, compared to non-trained group (Unpaired Student’s *t* test). All values are expressed as mean ± SEM.

**Table 1 animals-12-01759-t001:** Educational plan: activities and aims in chronological sequence.

Activity	Aim
Luring technique	Dogs are required to follow the human hand, which is considered a non-aversive target and/or movement to be done. Of note, in this way dogs experience positive emotions. The present activity underpins many others, involving control and direction signals.
Nose work activity	Aroused dogs learn to perform olfactory research following the verbal signal of “Search”, with the aim of lowering the state of emotional activation. Olfactory research improves the concentration attitude, and places the dog at the most appropriate emotional level for learning process.
The command “sit”	Dogs acquire a dog trainer-guided sitting posture, by means of non-verbal communication.
Management of the leash	Dogs learn to walk in a cooperative manner with the human, without tugging, nor pulling.
The command “stay”	Dogs are trained to be in awaiting state (no action required), as much as they can, and far from the trainer, until something will happen. The longer they stay the better they perform this activity.
Recall	Dogs learn to return to the human when recalled. They receive a treat, a cuddle, a compliment or start playing a game they like together with the trainer.
Feel safe during touch	Dogs are caressed in the direction of the hair, at first avoiding the head, tail and paws and preferring the trunk. The aim is to potentiate the pleasure of tactile contact, discovering where that subject prefers to be caressed.
How to improve intraspecific socialization	Dogs are required to experience pleasure when spending some time together with dogs housed in different boxes, modulating their impulsivity, aiming at mastering a good intraspecific communication. Dogs are chosen by their personalities and compatibility.
The command “give the paw”	Dogs learn to give the paw to the human, when required.

**Table 2 animals-12-01759-t002:** The number of dogs adopted between 2015 and 2020, subdivided by juvenile (7–11 months); young-adult (12–24 months); adult (3–8 years); senior (9–17 years). The dogs who entered the shelter between 2015 and 2017 did not perform the educational training (non-trained dogs). The animals analyzed between 2018 and 2020 were enrolled in the training program (Trained dogs).

Number of Dog Adoptions							
	Non-Trained Dogs			Trained Dogs			
Group	2015	2016	2017	2018	2019	2020	Total
Juvenile	142	106	57	73	64	125	567
Young-adult	42	42	22	38	50	61	255
Adult	21	13	22	36	48	30	170
Senior	3	2	7	10	12	8	42
Total	208	163	108	157	174	224	1034

## Data Availability

Not applicable.

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
