# Peer review of "The Potential Impact of a Dog Training Program on the Animal Adoptions in an Italian Shelter"

_animals, 2022, doi:10.3390/ani12141759_

Round 1

Reviewer 1 Report

Dear Authors,

Thank you for resubmitting this work, which I find much improved compared to first submission.

with kind regards

Author Response

We thank the reviewer for her/his relevant support in improving the quality of the new version of the manuscript.

Reviewer 2 Report

Comments on the manuscript “The potential impact of a dog training program on the animal adoptions in an Italian shelter” submitted to the Animals

General comments

This manuscript analyzes the potential effect of dog training in a kennel in southern Italy to increase adoption by people. It is a topic of interest to many professionals and the public looking to adopt a dog. I have high expectations for the manuscript, but there are methodological and structural problems that weaken the potential for publication. There is confusion between the methods section and the discussion, in addition to having two major focuses that do not match the objective proposed by the authors. One of the focuses is partially consistent with the objective, which is to address the crucial role of the educational positive reinforcement training upon behavioral functions in kenneled dogs. The other focus of the manuscript is the description of the structure and management of the kennel, discussing the economic advantages, sustainability, and welfare of the dogs. Scattered between these two major focuses, there are fragmentations that do not fit coherently with the main objective, such as the attitude to establish a more suitable human-dog interaction and dog welfare.

I agree that writing scientific articles can be emotional, but the expression of emotions is personal. It is not usual for qualifying and emotional expressions to appear in the text. The present text has some emotional nuances that do not seem suitable for a manuscript of this nature; therefore, it is strange to read in some sentences the words "sadly" (lines 53 and 261) and "tantalizing impact" (line 205). I suggest reviewing these terms for less subjective wording.

The paragraphs are very long, making them difficult to read. Ideally, paragraphs should be harmonic, not too short (single paragraph), and not too long. I suggest that writing needs to be improved by an experienced editing service.

In detail, I point out complimentary comments on the manuscript, as follows:

Line 36: There are no physiological data in the study. The conclusion of the abstract does not match what was observed.

Line 39: The study does not address dog wellbeing, thus, this word should be removed from indexers.

Lines 43 and 56: I do not understand how the expression "of course" is appropriate in the sentence.

Line 56: The “Law August 14, 1991…” must be written as a reference. Please read the instructions for the authors.

Line 64: Jakovcevic and collaborators (reference 12) wrote that "high sociability dogs gazed significantly longer at humans during extinction trials". It does not appear that gazing behavior “allows dogs to have a high level of sociability, interact with each other more easily, have positive interactions with strangers, and adapt to changing situations”.

Line 74: I don't understand why the authors refer to a study with horses when Ziv (2017) published a comprehensive review of positive and negative reinforcement in dogs.

Line 76: The authors refer to Vieira de Castro and collaborators as a single work, but refer to two articles with different authors (references 19 and 20). Is the argument based on one or two articles?

Line 85-88: The "crucial" role is a prejudgment of the results. If the role of training is “crucial”, why do this study, which is to demonstrate how training facilitates the action of dogs? In this same line of thought, the authors mention that "positive education" enhances the attitude to establish a more suitable human-dog interaction. I agree that adoption can be a strong indication of a more suitable human-dog interaction, but not just that. There are other competing motivations for wanting to adopt a dog.

Line 98-100: Need to improve this sentence.

Line: Reference 23 deals with an analysis of cortisol in sheltered dogs in a program of animal-assisted interventions in a prison. If the purpose of the sentence is to show the reader that the kennel has already been described in another article, this should add an explicit sentence (e.g. "further details of the kennel have already been described in d’Angelo et al , 2021")

Line 106-122: This information is not linked to the objective, methods, and results of the study. This information is not relevant.

Line 112: “(https://www.isprambiente.gov.it/it/attivita/certificazioni/emas/il-regolamento-emas)” must be written as a reference. Please read the instructions for the authors.

Line 127: It is not usual to use the term "adolescent" for the age class of dogs. More common is to use the terms infant, juvenile, adult (and sub-adult), and senior. Please review the scientific literature.

Line 132: What do you mean that dog training can be useful for “zooanthropology with child”?

Line 134: I understand that a “behaviorist” is a professional who applies Behaviorism theory and methods. Behaviorism theory of learning is based on the idea that all behaviors are acquired through conditioning, and conditioning occurs through interaction with the environment. Is this the methodological line of action of the veterinarian involved in the study?

Line 153: The legend must go above table 1. Inside the table, there is a description of the “stay” command, in which the dog must be in a "non-physiological posture". How can a dog be in a non-physiological posture? Strictly speaking, non-physiological is a clinical position, that is, with pathological changes. Therefore, the "stay" command needs to be redefined.

Line: Four age classes make the comparison between trained and untrained dogs. The comparison was by the absolute numbers, that is, without the power of the sample by age class, resulting in a bias in the analysis. If we transform the data into percentages (table 1), it is observed that the classes of trained and untrained pre-teen and adolescent dogs are balanced (53.95% vs 46%; 41.62% vs 58.4%). However, this balance does not occur between the adult and old (elderly) age classes. Adult trained dogs are 67% compared to 32.84% of untrained dogs. Old trained dogs are 71.4% of the sample compared to 28.6% of untrained old dogs. Not surprisingly, in the adult and senior categories, there were more adoptions, because there are more animals available for adoption. Therefore, there is no distinction between the purely numerical effect of available quantity and the training effect. The data must be weighted so that the effect (if any) of training on adoption is observable. All data supporting the arguments of the discussion and conclusion are committed to this failure.

Table 1: sample size data in percentages of dogs by age class, trained, and untrained.

Untrained

Trained dogs

Pre adolescent

54%

46%

Post adolescent

42%

58%

Adult

33%

67%

Old (elderly)

29%

71%

Line 182: The three asterisks are missing to indicate P<0.001.

Line 186-189: This grouping for analysis has already been described in the methods.

Line 202-307: There are several arguments that do not relate to the focus of the study, which are training/non-training and adoption/non-adoption. Furthermore, as mentioned above, discussion and conclusion are hampered by bias in data analysis.

Reference

Ziv, G. (2017). The effects of using aversive training methods in dogs—A review. Journal of Veterinary Behavior, 19, 50-60.

Author Response

D’Angelo et al. 2022-Animals-1766547-Rebuttal

Reviewer 2

General comments

This manuscript analyzes the potential effect of dog training in a kennel in southern Italy to increase adoption by people. It is a topic of interest to many professionals and the public looking to adopt a dog. I have high expectations for the manuscript, but there are methodological and structural problems that weaken the potential for publication. There is confusion between the methods section and the discussion, in addition to having two major focuses that do not match the objective proposed by the authors. One of the focuses is partially consistent with the objective, which is to address the crucial role of the educational positive reinforcement training upon behavioral functions in kenneled dogs. The other focus of the manuscript is the description of the structure and management of the kennel, discussing the economic advantages, sustainability, and welfare of the dogs. Scattered between these two major focuses, there are fragmentations that do not fit coherently with the main objective, such as the attitude to establish a more suitable human-dog interaction and dog welfare.

I agree that writing scientific articles can be emotional, but the expression of emotions is personal. It is not usual for qualifying and emotional expressions to appear in the text. The present text has some emotional nuances that do not seem suitable for a manuscript of this nature; therefore, it is strange to read in some sentences the words "sadly" (lines 53 and 261) and "tantalizing impact" (line 205). I suggest reviewing these terms for less subjective wording.

The paragraphs are very long, making them difficult to read. Ideally, paragraphs should be harmonic, not too short (single paragraph), and not too long. I suggest that writing needs to be improved by an experienced editing service.

  1. We thank the reviewer for Her/His thoughtful comments, that allowed us to definitely improve the quality of the work. In the revised version of the manuscript, we sought to draw attention just on the potential influence of the training upon the chances of adoption and, hopefully, which may pave the way for a better human-dog relationship. To accomplish this, we used a biased approach, considering the number of adoptions of trained dogs within each age group, revealing that adult and senior trained dogs were more prone to be adopted. In this respect, as discussed in the main text, we found a significantly higher number of total adoptions in the trained dogs, compared to the control animals, since over 735 trained dogs, 555 were adopted, instead of 479 out of 1387 non-trained animals (chi-square test: df = 322.9, 1; z = 17,97; p < 0.0001). However, this apparently amazing data should be considered very carefully, since adoption is a complex phenomenon, affected by many factors, besides training, to take into consideration, including appearance, social interaction with the adopter and personality. We added this new section in the discussion. Finally, we sought to smooth expressions to make the paper less emotional, and amended all the text according to your kind suggestions.

In detail, I point out complimentary comments on the manuscript, as follows:

Line 36: There are no physiological data in the study. The conclusion of the abstract does not match what was observed.

  1. We amended this point.

Line 39: The study does not address dog wellbeing; thus, this word should be removed from indexers.

  1. We removed that word.

Lines 43 and 56: I do not understand how the expression "of course" is appropriate in the sentence.

  1. We removed this expression.

Line 56: The “Law August 14, 1991…” must be written as a reference. Please read the instructions for the authors.

  1. We did it.

Line 64: Jakovcevic and collaborators (reference 12) wrote that "high sociability dogs gazed significantly longer at humans during extinction trials". It does not appear that gazing behavior “allows dogs to have a high level of sociability, interact with each other more easily, have positive interactions with strangers, and adapt to changing situations”.

  1. We amended this point.

Line 74: I don't understand why the authors refer to a study with horses when Ziv (2017) published a comprehensive review of positive and negative reinforcement in dogs.

  1. We amended this point and added the suggested reference.

Line 76: The authors refer to Vieira de Castro and collaborators as a single work, but refer to two articles with different authors (references 19 and 20). Is the argument based on one or two articles?

  1. We removed ref 20, and left the more appropriate one (19).

Line 85-88: The "crucial" role is a prejudgment of the results. If the role of training is “crucial”, why do this study, which is to demonstrate how training facilitates the action of dogs? In this same line of thought, the authors mention that "positive education" enhances the attitude to establish a more suitable human-dog interaction. I agree that adoption can be a strong indication of a more suitable human-dog interaction, but not just that. There are other competing motivations for wanting to adopt a dog.

  1. We smoothed this point.

Line 98-100: Need to improve this sentence.

  1. We did it.

Line: Reference 23 deals with an analysis of cortisol in sheltered dogs in a program of animal-assisted interventions in a prison. If the purpose of the sentence is to show the reader that the kennel has already been described in another article, this should add an explicit sentence (e.g. "further details of the kennel have already been described in d’Angelo et al , 2021")

  1. We did it.

Line 106-122: This information is not linked to the objective, methods, and results of the study. This information is not relevant.

  1. We did it.

Line 112: “(https://www.isprambiente.gov.it/it/attivita/certificazioni/emas/il-regolamento-emas)” must be written as a reference. Please read the instructions for the authors.

  1. We did it.

Line 127: It is not usual to use the term "adolescent" for the age class of dogs. More common is to use the terms infant, juvenile, adult (and sub-adult), and senior. Please review the scientific literature.

  1. We did it. In the revised version of the manuscript, we used the following terms: juvenile, young-adult, adult, senior, according to what indicated by Harvey et al., 2021.

Line 132: What do you mean that dog training can be useful for “zooanthropology with child”?

  1. We understand the potentially misleading concept, so we removed the section concerning the zooanthropology with child. In this respect zooanthropology is discipline that studies the human-animal relationship and yes aims to improve the approach with the animal and integration of the animal in the scope social and family; use the relationship with the animal to get positive effects on humans.

Line 134: I understand that a “behaviorist” is a professional who applies Behaviorism theory and methods. Behaviorism theory of learning is based on the idea that all behaviors are acquired through conditioning, and conditioning occurs through interaction with the environment. Is this the methodological line of action of the veterinarian involved in the study?

  1. We thank the referee for her/his comment. However, in Italy, the Veterinary Behaviorist is generally regarded as a professional with know-how on Behavioral Medicine, Internal Medicine and Clinical Ethology. Therefore, the behavioral veterinarian in the present research employed a systemic-integrated approach (see d'Angelo D, Sacchettino L, Carpentieri R, Avallone L, Gatta C, Napolitano F. An Interdisciplinary Approach for Compulsive Behavior in Dogs: A Case Report. Front Vet Sci. 2022 Mar 24;9:801636. doi: 10.3389/fvets.2022.801636), based not only upon behavior attitudes, but also cognition, emotions, relationship with others and individual previous experiences.

Line 153: The legend must go above table 1. Inside the table, there is a description of the “stay” command, in which the dog must be in a "non-physiological posture". How can a dog be in a non-physiological posture? Strictly speaking, non-physiological is a clinical position, that is, with pathological changes. Therefore, the "stay" command needs to be redefined.

  1. We understand the reviewer. We used the "non-physiological" term to refer to the will of dogs to act, to be physiologically a species aimed at action, rather than being static, unmoving. However, thanks to the advice, we have rephrased the sentence. We moved the legend above the table 1.

Line: Four age classes make the comparison between trained and untrained dogs. The comparison was by the absolute numbers, that is, without the power of the sample by age class, resulting in a bias in the analysis. If we transform the data into percentages (table 1), it is observed that the classes of trained and untrained pre-teen and adolescent dogs are balanced (53.95% vs 46%; 41.62% vs 58.4%). However, this balance does not occur between the adult and old (elderly) age classes. Adult trained dogs are 67% compared to 32.84% of untrained dogs. Old trained dogs are 71.4% of the sample compared to 28.6% of untrained old dogs. Not surprisingly, in the adult and senior categories, there were more adoptions, because there are more animals available for adoption. Therefore, there is no distinction between the purely numerical effect of available quantity and the training effect. The data must be weighted so that the effect (if any) of training on adoption is observable. All data supporting the arguments of the discussion and conclusion are committed to this failure.

Table 1: sample size data in percentages of dogs by age class, trained, and untrained.

Untrained

Trained dogs

Pre adolescent

54%

46%

Post adolescent

42%

58%

Adult

33%

67%

Old (elderly)

29%

71%

  1. In the previous version we did not consider the global effect of the training program on the adoption, because the strong effect we observed (75 % of trained dogs adopted, compared to 34 % of non-trained dogs, independently on the age) was likely to be misleading, given the complexity of adoption phenomenon. On the other hand, having found a sort of turnaround, with adult and senior trained dogs more prone to be adopted, when compared to the age-matched controls, suggest that training may reveal beneficial for a such purpose. See the above (general comment) for further considerations.

Line 182: The three asterisks are missing to indicate P<0.001.

  1. We did it.

Line 186-189: This grouping for analysis has already been described in the methods.

  1. We did it.

Line 202-307: There are several arguments that do not relate to the focus of the study, which are training/non-training and adoption/non-adoption. Furthermore, as mentioned above, discussion and conclusion are hampered by bias in data analysis.

  1. As reported above in the general comments, in the revised version of the manuscript we tried to focus just on the influence of the training on the adoption chances, in an age-related manner.

Round 2

Reviewer 2 Report

Dear editor

Thank you for the opportunity to comment on this manuscript. The current version is heavily modified. The authors did a good job of revising the previous version. There were substantial improvements that made the manuscript more coherent and readable. The text is innovative and provocative because it could raise debates about the training of dogs in shelters facilitating adoption. The manuscript is suitable for publication.

Author Response

Thank the reviewer for her/his consideration.

This manuscript is a resubmission of an earlier submission. The following is a list of the peer review reports and author responses from that submission.

Round 1

Reviewer 1 Report

Review Notes:

It is clear the authors are interested in promoting a more holistic view of one-health in animal shelters, which is commendable. 

Grammatical Comments-

There are some typos (such as double commas in the summary) throughout the paper.  I would suggest the authors go through the paper carefully looking for typos and language issues.  In addition, I would remove any unnecessary repetitive language (such as interestingly in the simple summary).  Terminology issues include using “imprisoned” for dogs confinement.  The description of the Italian law is also very unclear (line 59-61). 

One-Health View-

I think that “one-health” is not really the right context or umbrella to coordinate this paper.  I believe it should be dropped entirely and reframed.   I am therefore going to reject the paper so it can be rewritten and submitted without the one-health information that isn’t relevant to their project.  I feel like it is misleading and using a buzz word to encapsulate something that is not significant in the research conducted.  Also, much more substantial discussion is needed in this paper.  I believe a paper rewritten as “The potential impact of a dog training program on animal adoptions in an Italian shelter” is a much more accurate description.  The environmental pieces here are superfluous and seem to be included just to meet a “quota” of one-health. 

Research methodology-

The way the research methodology is described isn’t very helpful for extrapolating information.  The training program information is interesting, but training sessions seem long and it seems like this was just used for selection.  More clear hypotheses, methodology, and distinct discussion of the purpose of the study is needed.  I would rewrite this paper to focus on the dog training piece and drop the one-health context. 

Reviewer 2 Report

Dear authors,

General comments

I think that this work is interesting and on a relevant topic. However, in my opinion considerable improvement is needed prior to reconsider this work for publication in “Animals”.

Statistical analysis: this paragraph should be more detailed and the information regarding the statistical tests used should not be repeated elsewhere (as it is the case in paragraph 3.3). It is not clear how ANOVA was used to analyse the number of adoptions based on age. ANOVA is a test used to assess a difference in means between multiple (>2) treatment groups, and data should include one categorical independent variable (e.g., a treatment) and one quantitative (normally distributed) dependent variable.  If the age was used as “treatment”, then I presume that the number of dogs adopted within the two periods of time (2015-2017 and 2018-2020) would be your quantitative variable; however, these are numbers (one number per age group/period) and not means, which breaks the assumptions for using ANOVA. A better approach to look at this set of data could have been using proportions instead of number of dogs. Please, consult a statistician regarding the statistical methods more appropriate for your set of data and review data analysis accordingly. Also, at paragraph 3.4 it is not clear how the number of dogs was used as quantitative dependent variables for two-way ANOVA. Can the authors please provide further explanations on the statistical methods used?

I think that the technical description of the kennel is interesting and potentially useful to be used as a model of modern and functional kennel. However, it is a bit difficult to link this to the main focus of the paper. I think it would be useful to emphasise somewhere in the manuscript (e.g., in the introduction) how the quality of the infrastructures/facility would contribute to the wellbeing of the dogs.

Kind regards.